# Flowering time and seed dormancy control use external coincidence to generate life history strategy

Vicki Springthorpe[1], Steven Penfield[2]*

[1]Department of Biology, University of York, York, United Kingdom; [2]Department of Crop Genetics, John Innes Centre, Norwich, United Kingdom

**Abstract** Climate change is accelerating plant developmental transitions coordinated with the seasons in temperate environments. To understand the importance of these timing advances for a stable life history strategy, we constructed a full life cycle model of *Arabidopsis thaliana*. Modelling and field data reveal that a cryptic function of flowering time control is to limit seed set of winter annuals to an ambient temperature window which coincides with a temperature-sensitive switch in seed dormancy state. This coincidence is predicted to be conserved independent of climate at the expense of flowering date, suggesting that temperature control of flowering time has evolved to constrain seed set environment and therefore frequency of dormant and non-dormant seed states. We show that late flowering can disrupt this bet-hedging germination strategy. Our analysis shows that life history modelling can reveal hidden fitness constraints and identify non-obvious selection pressures as emergent features.

## Introduction

Study of climate effects on phenology have quantified shifts in the timing of developmental transitions such as flowering, bud burst, and migration. These show that warmer temperatures are advancing plant phenology, with bud burst and flowering occurring earlier and the growing season becoming longer (*Menzel and Fabian, 1999*; *Fitter and Fitter, 2002*; *Cotton, 2003*; *Parmesan and Yohe, 2003*; *Cleland et al., 2007*). This is an adaptive plant response to shifting temperatures, because for many species the flowering date is unaffected by climate change (*Fitter and Fitter, 2002*), and plants that couple their phenology to temperature are more likely to have populations resilient to climate change (*Willis et al., 2008*). Why many plants have evolved to behave in this manner is currently unclear.

The model plant *Arabidopsis thaliana* is widely distributed in the Northern hemisphere and can be used to analyze phenology, including control by individual genes (*Wilczek et al., 2009*; *Chiang et al., 2013*). In northern Europe, *Arabidopsis* exhibits a primarily winter annual life history, but in central and southern Europe, summer rapid cycling and summer annuals appear alongside winter annuals (*Thompson, 1994*). Importantly, *A. thaliana* is a ruderal species, primarily colonizing frequently disturbed habitats: the long-term persistence of such populations (and therefore fitness) is strongly dependent on seed behaviour and seed bank formation (*Grime et al., 1981*).

During the vegetative phase, plants have signalling pathways that couple progression to flowering to temperature and photoperiodic cues, and these pathways have been elucidated genetically under laboratory conditions (*Andrés and Coupland, 2012*). These same pathways are subject to positive selection in natural populations (*Toomajian et al., 2006*). However, when late flowering *Arabidopsis* mutants are analysed under field conditions they delay reproduction by only a few days, leaving the significance or their role unclear (*Wilczek and et al., 2009*; *Chiang et al., 2013*).

**\*For correspondence:** steven. penfield@jic.ac.uk

**eLife digest** Plants adjust when they grow, develop flowers and produce, or 'set', seeds in response to changes in temperature and day length. It is therefore unsurprising that climate change alters the timing of these important events in plants' lives; for example, many plants are adapting to rising temperatures by flowering earlier and growing for longer.

The environmental signals that control when a plant flowers, and the genes that underlie this process, have been well studied in the model plant *Arabidopsis thaliana*. This plant's ability to quickly colonize and thrive in disturbed habitats—including agricultural land, construction sites, and waste ground—is partly because some of its seeds lie dormant in the soil, for up to several years, before they start to grow. Whether or not a seed undergoes a period of dormancy is controlled by the temperature that the seeds experienced when they were developing; this in turn is influenced by earlier events, such as when the flowers first developed, and when the plant first started to grow from its seed (a process called germination).

To try to understand these complex interactions, Springthorpe and Penfield developed a computational model of the major events in the life of an *Arabidopsis* plant. Data collected from *Arabidopsis* plants that normally germinate in winter and spring were then used to check whether the model could accurately represent what happens in nature.

The analysis confirmed that the timing of seed setting depends mostly on the environmental temperature. Springthorpe and Penfield then showed that plants both flowered and set seed earlier in response to increases in temperature, so that the seeds were shed precisely when the temperature was between 14°C and 15°C.

Springthorpe and Penfield discovered that rise in the average temperature when a plant set seed from 14°C to 15°C had a dramatic effect on the seeds. Almost all of the seeds that developed below 14°C became dormant, while very few of the seeds that developed above 15°C became dormant.

From their findings, Springthorpe and Penfield predict that the temperature control of flowering time has evolved to constrain when seeds are set and ensure that plants produce a mixture of seeds: some that will become dormant, and some that will not. Their findings also show that modelling the whole life history of an organism has the potential to reveal strategies that are not obvious when studying single events in isolation. If the model was extended to include genetic variation across populations of plants, this approach could give new insights into how individual genes help plants adapt to weather and climate.

Temperature during seed set also strongly influences life history by modulating seed dormancy after shedding (*Fenner, 1991*; *Schmuths et al., 2006*; *Chiang et al., 2009*; *Kendall et al., 2011*), but in contrast to the control of flowering time comparatively little attention has been paid to the role of this process in life history generation and adaptation. Given that the environment during seed set is determined by flowering time, we hypothesized that understanding the role of either flowering time or dormancy control in population life history required a detailed integrated study of both traits. To address this problem, we therefore constructed and parameterized a field-validated model of *Arabidopsis* life history for the Columbia-0 (Col-0) accession.

## Results

Previously, photothermal models have been used to predict *Arabidopsis* flowering time under field conditions (*Wilczek et al., 2009*). In order to predict seed set conditions in the wild, we sought to extend this treatment to include the reproductive phase in order to determine the timing of seed set on plants germinated and grown at different times of the year. To parameterize this model, we first grew plants from bolting to first set seed under a range of temperatures under laboratory conditions (*Figure 1A*). We found a simple linear relationship between temperature and the number of days from bolting to first seed set, showing that seed set timing depended on ambient temperature. Our data suggested that seed set was much less dependent on photoperiod, with only photoperiods of 8 hr significantly delaying seed set (*Figure 1—figure supplement 1*). With these data, we constructed and optimized a thermal time model from first flowering to first seed set, and compared the predicted timing of seed set to that of plants setting seed under field conditions in York, UK in 2012 and 2013

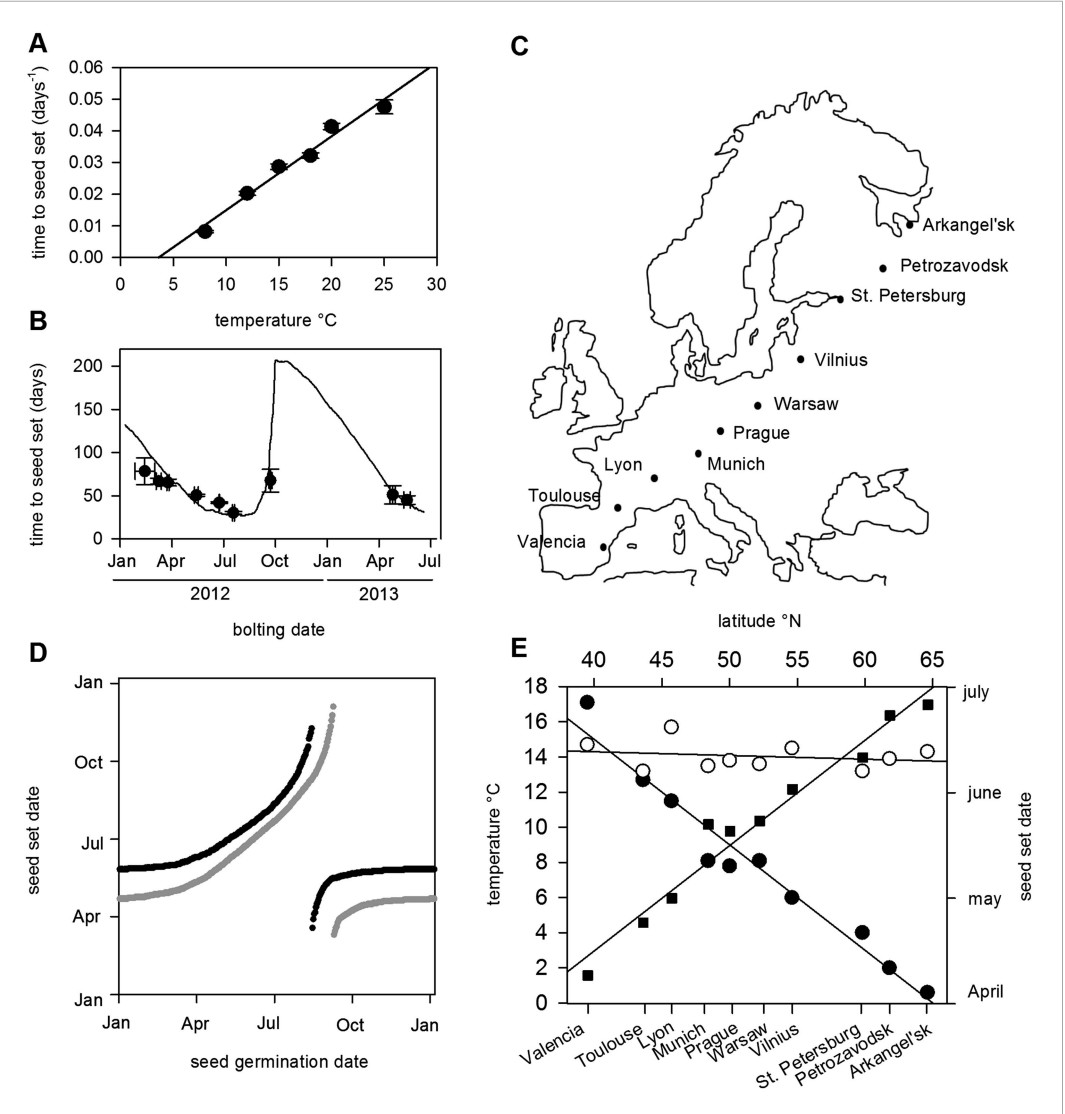

**Figure 1**. Flowering time and seed set control constrain mean temperature at seed shedding for winter and spring annuals. (**A**) Laboratory experiments from bolting to first seed set at various constant temperatures used to constrain the seed set thermal time model. Data represent the mean and standard error of a minimum of five independent plants per treatment. (**B**) Field trial testing of the seed set model using nine growings of a minimum of five plants showing mean and standard error of bolting date and time to seed set. Closed circles show data while model prediction is shown by the continuous line. (**C**) Map of Central Europe showing the sites for which temperature data was gathered to simulate Col-0 behaviour. (**D**) Simulation of flowering time (grey) using the model from (*Wilczek and et al., 2009*) and first seed set model (black) using temperature data from Gorsow, Poland. Flowering and first seed set date (y-axis) can be determined for each possible germination date (x-axis). Note that seed set dates are similar for germination between September and April. (**E**) Predicted first seed set date (closed squares), predicted mean temperature at first seed set (open circles), and mean annual temperature (closed circles) for October/November germination dates at each site. Seed set is shown to be clustered around a mean daily temperature of 14.5°C.

The following figure supplement is available for figure 1:

**Figure supplement 1**. The growth rate of *Arabidopsis* Col-0 plant between first flowering and first seed set at photoperiods between 8 and 16 hr daylength at 22°C.

(*Figure 1B*). The model was a good predictor of seed set timing in the field ($R^2 = 0.43$) for seed set in York in 2012 and 2013, and outperformed models that also included photoperiod components. By combining this with a previously published bolting time model (*Wilczek et al., 2009*), we could predict Col-0 first seed set date for any germination date.

The precise location for the collection of Col-0 is uncertain but it is known to originate from central Europe (*Robbelen, 1965*; C Dean Personal communication). Simulated at the most likely collection site, Gorsow, Poland, a key feature of the model is that germination times from September to April (winter and spring annuals) lead to a very similar seed set time in late May (*Figure 1C*), a behaviour caused in part by growth rate and flowering time control (*Wilczek and et al., 2009*) but further enhanced by the seed set responses to temperature. Seed germinating from May to July set seed later in the same summer (rapid cycle), whereas August germinators have a wide range of possible flowering times (*Wilczek and et al., 2009*). To understand how this would vary in response to shifting climate norms, we simulated the flowering and seed set models using climate data across a transect covering 25° of latitude from Arkangel'sk, Russia, to Valencia, Spain (*Figure 1D,E*). Warmer climates advanced the seed set date, with 1°C rises in annual temperature giving about a 7-day advance in seed set time for winter annual cohorts from late June in Arkhangel'sk to early April in Valencia (*Figure 1E*). This behaviour has been widely observed in phenological studies, but our analysis also revealed a striking effect on temperature at first seed set: although mean annual temperature at the sites varied between 0.5°C and 17.5°C, winter annual seed set temperature was maintained at 14–15°C independent of latitude (*Figure 1E*). Therefore, a key emergent feature of the model is that the acceleration of flowering and seed set timing by warmer temperatures serves to stabilize the temperature range during which seed set begins.

Lower temperatures during seed production enhance seed dormancy in most angiosperm species (*Fenner, 1991*), but rarely has this response been characterized in detail. Laboratory experiments revealed that Col-0 dormancy appeared related to the mean daily temperature and was not clearly influenced by daily temperature cycles (*Figure 2—figure supplement 1*). To understand precisely how Col-0 seed dormancy responds to temperature during seed set, we matured seeds between 10°C and 20°C in the laboratory and germinated them following incubation treatments at temperatures between 4°C and 20°C (*Figure 2*). Our results revealed a clear bi-phasic behaviour: At or below 14°C, seeds produced are very dormant and do not germinate at high levels even after prolonged incubations under any temperatures. This is not due to low viability because seeds set at 10°C are viable at harvest (*Kendall et al., 2011*). In contrast, above 14°C, high levels of germination were possible especially where seeds were incubated at similar or lower temperatures than experienced during seed set. At these temperatures, seeds incubated in the dark experienced a transient period of primary dormancy loss during which time germination could occur if the seeds were given light, followed by secondary dormancy induction. This high sensitivity of dormancy to a 1°C increase in seed maturation temperature supports our previous observation of a general increase in the temperature-sensitivity of gene expression in developing seeds relative to vegetative tissues (*Kendall et al., 2011*).

To allow simulation of the whole life history, we developed a predictive framework for Col-0 germination probability and constructed a germination model based on the following assumptions used previously for similar studies (*Totterdell and Roberts, 1979*; *Bradford, 2005*; *Batlla et al., 2009*): (1) germination is possible in the absence of dormancy; (2) primary dormancy depth (dormancy generated during seed maturation) is fixed at harvest and is lost during imbibition; (3) the onset of secondary dormancy (dormancy induced de novo after imbibition) begins independently during seed imbibition. Primary dormancy loss and secondary dormancy induction were represented as cumulative logistic functions (see 'Materials and methods'). We noted that secondary dormancy appeared to be induced faster at warmer imbibition temperatures (*Figure 2*), and fixing this relationship in the model revealed that close matches to experimental data could be produced if initial primary dormancy was modelled as a function of the temperature during seed maturation (see 'Materials and methods' and *Figure 2—figure supplements 2–4*). In fact, a model with four parameters in which primary dormancy loss was dependent linearly on maturation temperature and secondary dormancy induction was dependent exponentially on imbibition temperature could be parameterised to provide a good fit to experimental data generated between 12 and 20°C ($R^2 = 0.84$; *Figure 2—figure supplement 2*) and could match germination data in an experimental series set at 16°C and not used to train the model (*Figure 2—figure supplement 3*). In order to determine whether our model could predict the germination of seed set under field conditions, we collected five seed batches during 2012 in a field

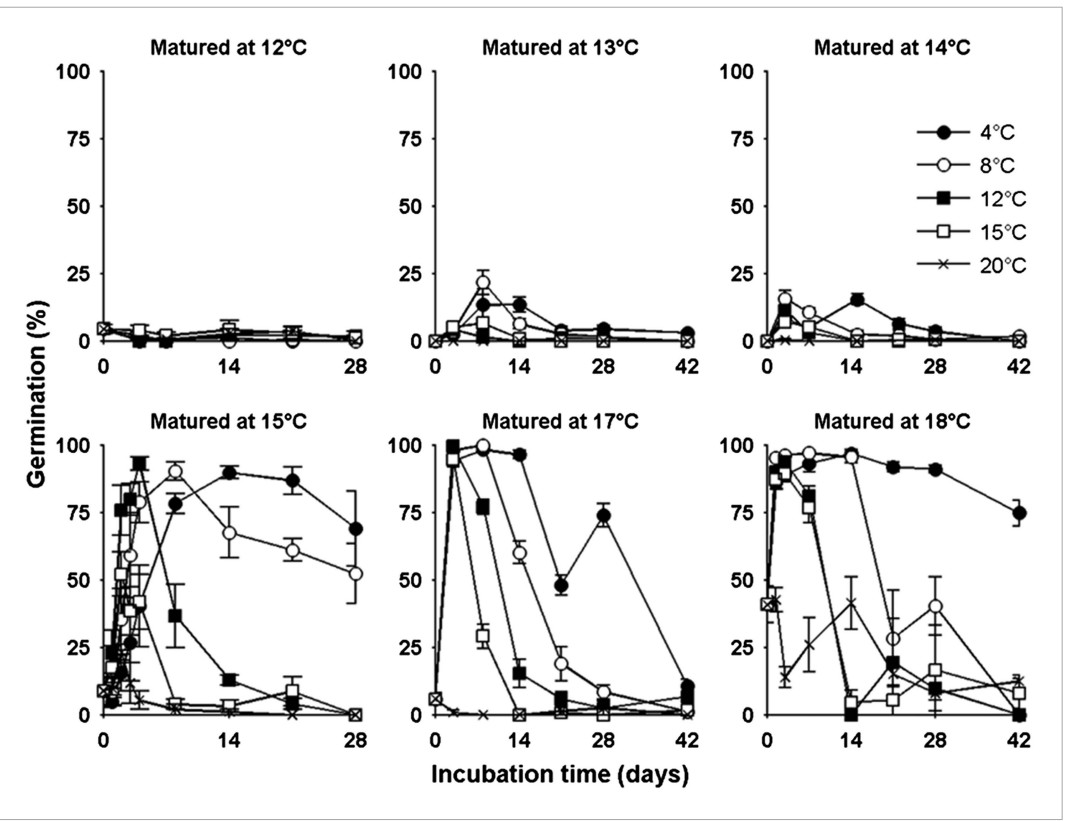

**Figure 2**. The germination physiology of *Arabidopsis* Col-0 in response to temperature during seed set and imbibition. Germination of seed matured between 12°C and 18°C incubated in the dark between 4°C and 20°C (see legend) and placed to germinate in the light at 22°C after the indicated time periods. Data points represent mean and standard error of five seed batches.

The following figure supplements are available for figure 2:

**Figure supplement 1**. Comparison of the relative effects of constant 16°C vs a 24-hr temperature cycle with a mean of 16°C on Col-0 seed dormancy.

**Figure supplement 2**. Global fit of the seed germination model output (red) with the time-series training data (blue) for all seed maturation temperatures (Tm) and imbibition temperatures (Ti).

**Figure supplement 3**. Fit of model to lab-generated data not used to train the model.

**Figure supplement 4**. Germination kinetics of field-collected seed batches from 2012 and model prediction of temperature during seed maturation.

site in York, UK and germinated them under multiple temperature regimes in the laboratory (*Figure 2—figure supplement 4*). Behaviour of these lots was qualitatively similar to that observed in the laboratory. We then fitted the parameterised model to each field germination data set, generating a prediction for the seed maturation temperature for each seed lot. This was then compared to the temperature history of each batch in the field (*Figure 2—figure supplement 4*). In each case the model-predicted temperature was close to the daily mean temperature in the last days before harvest for each seed batch, as has been shown previously in barley (*Gualano and Benech-Arnold, 2009*). Therefore, our model captures field-relevant relationships between maturation temperature and germination behaviour.

To understand the germination potential of seeds set at different times of the year, we ran the model for the locations in our transect (*Figure 3*). The germination model predicts behaviour with two

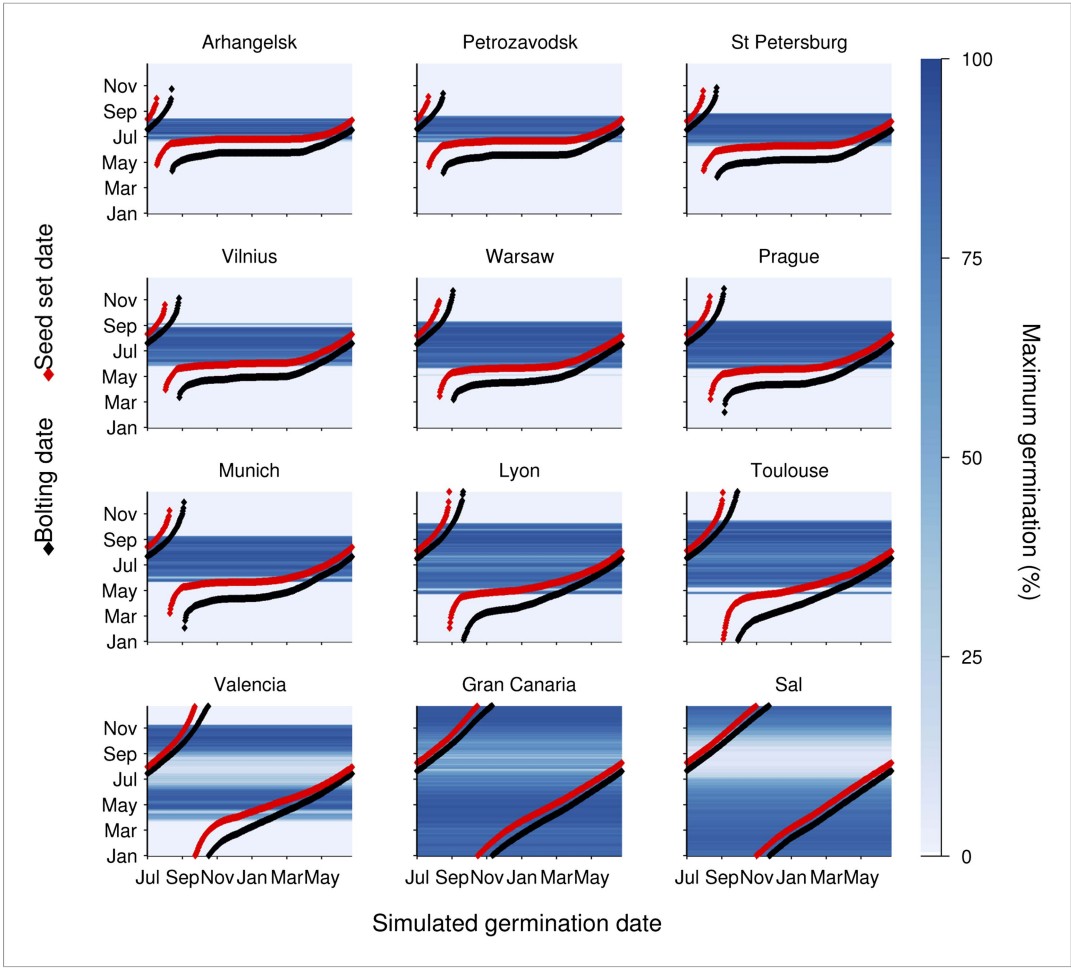

**Figure 3**. Simulation of the interaction of Col-0 life history stages at sub-arctic to sub-tropical sites. Germination dates are given on the x-axis. On the y-axis flowering time (red line) and first seed set (black line) are shown alongside the simulated progeny germination frequency (blue-hued heat map) based on the mean temperature over 1 week before shedding (seed maturation temperature) and temperature after shedding (imbibition temperature). Behaviour of winter annuals can be observed using September–November germination dates, and ability to complete a summer rapid cycle can be ascertained by progression to seed set in the same year for summer germination dates. The model is simulated for European locations on the transect introduced in *Figure 1*, and for the more sub-tropical locations of the Canary Islands and Cape Verde (Sal), both locations at which *Arabidopsis thaliana* populations have been described.

The following figure supplement is available for figure 3:

**Figure supplement 1**. The coincidence between seed set timing and the dormancy state transition is preserved during artificial warming and cooling simulations, based on 2°C increments from the mean temperature series in Gorsow, Poland.

principle states obvious from the data: Firstly, mean temperatures below 14°C during seed set strongly inhibit germination. Secondly, above 15°C the interaction between maturation temperature and imbibition temperature generally permits high germination rates. This is because faster gain of secondary dormancy after imbibition at warmer temperatures is offset by the lower level of primary dormancy induced during seed maturation. Warmer climates have a longer summer window of high germination until at very warm locations the model predicts germination inhibition at the height of summer, due to fast secondary dormancy induction. Our models show that the predicted transition from dormancy to germination takes place as the seed set temperature rises above 14°C, coinciding with the time of first seed set for winter annual cohorts, regardless of location (*Figure 3*). Therefore,

simulated Col-0 phenology generates a coincidence of seed set timing with germination physiology which is independent of climate over 25° of latitude, demonstrating that this aspect of Col-0 phenology is strongly buffered against variation in temperature. Our model predicts that the low dormant progeny will enter a rapid cycle if the climate permits, flowering and setting seed later the same summer (*Figure 3*). This switch therefore has the potential to control the proportion of seeds entering the seed bank and those that emerge immediately for a rapid cycle generation for production of further seeds.

Given the high sensitivity of Col-0 seed behaviour to temperature (*Figure 2*), we hypothesized that moderately delayed flowering could affect progeny life histories. To test this hypothesis, we ran the model using previously published parameter sets for simulating the *gigantea* (*gi*) and *vernalisation insensitive 3* (*vin3*) (*Sung and Amasino, 2004*; *Toomajian et al., 2006*; *Wilczek and et al., 2009*) for Gorsow, Poland (*Figure 4*), although in principle our model behaviour is relevant to large areas of central Europe (*Figure 1*; *Figure 3*). GI confers a weak late flowering in field conditions of 7–10 days during summer while *vin3-1* mutants generate a longer delay of 25–50 days (*Sung and Amasino, 2004*; *Wilczek and et al., 2009*). Thus, these are representative of any natural variants with mild and more severe delays in flowering affecting temperature or photoperiod sensitivity. This is more informative than considering strong *FRI* alleles because these do not substantially delay flowering of winter annuals (*Wilczek and et al., 2009*; *Chiang et al., 2013*). Both *gi* and *vin3* mutants were predicted to set seed later in the year than wild type (*Wilczek and et al., 2009*), producing a greater proportion of seeds above the 14.5°C threshold and theoretically giving rise to more seeds capable of immediate germination. Given the time of year, these germinating seeds would be committed to a rapid cycling life history (*Figure 1D*). Assuming that seed set lasts for 1 month and follows a normal distribution, simulated across our transect the model predicts that *gi* mutations could double the number of low-dormant seeds in northern and central Europe but would have little effect at lower latitudes (*Figure 4B*). Mean temperature at shedding was raised from 14°C to 15°C for *gi* mutants at sensitive sites (*Figure 4A,B*). For *vin3* the model predicts a rise in the mean temperature at maturation of almost 5°C, enough to commit most seeds to rapid cycling and severely curtail entry into the soil seed bank (*Figure 4C*). Our analysis therefore suggests that small variations in flowering time may have dramatic effects on progeny life history in *Arabidopsis* by affecting seed dormancy, providing evidence for a previously unrecognized mechanism through which flowering time can affect plant fitness. This conclusion is supported by a recent study that shows that changing flowering time affects seed dormancy under field conditions more than flowering time itself (*Chiang et al., 2013*).

## Discussion

We have revealed that *Arabidopsis* Col-0 seed dormancy is remarkably sensitive to a 1°C rise in mean temperature during seed set from 14°C to 15°C (*Figure 2*). However, until now the significance of the response of primary dormancy to the temperature during seed maturation has not been understood. A central feature of *Arabidopsis* phenology is the ability to generate genetically identical cohorts of seeds with contrasting life histories, such as winter annuals and rapid cyclers (*Montesinos-Navarro et al., 2012*). Our data and models show that in a striking coincidence winter annual seeds are set when the mean temperature is approximately 14–15°C, the temperature at which a major switch from the production of dormant to non-dormant seeds occurs. This inevitably divides progeny into at least two distinct cohorts: obligate germinators which become summer rapid cyclers, and seeds which enter the soil seed bank even if initially exposed to light (*Figure 5*). Notably, a similar transition has been observed in *Capsella bursa-pastoris* seeds: in this case there is a transition in seed coat colour during seed set that alters dormancy and correlates with flowering time (*Toorop et al., 2012*). In *Arabidopsis* seed coat, pigmentation is affected by the temperature during seed set (*MacGregor et al., 2015*), again demonstrating control by the mother plant. Similar bet-hedging germination strategies have been described frequently across flowering plants suggesting wide general relevance (*Childs et al., 2010*), but further studies are required to understand whether this is a common strategy for *Arabidopsis* in locations where temperatures are warm enough to permit life history variation. In addition, our study only considers annual temperature and photoperiod variation and other factors, notably water availability, are also known to strongly affect germination and growth rates.

Because seed bank loading is a key component of fitness in ruderal plant populations (*Grime, 1988*), control of progeny behaviour likely generates a major selection pressure on flowering time control. Genetic analyses of *Arabidopsis* fitness have frequently considered yield to be paramount,

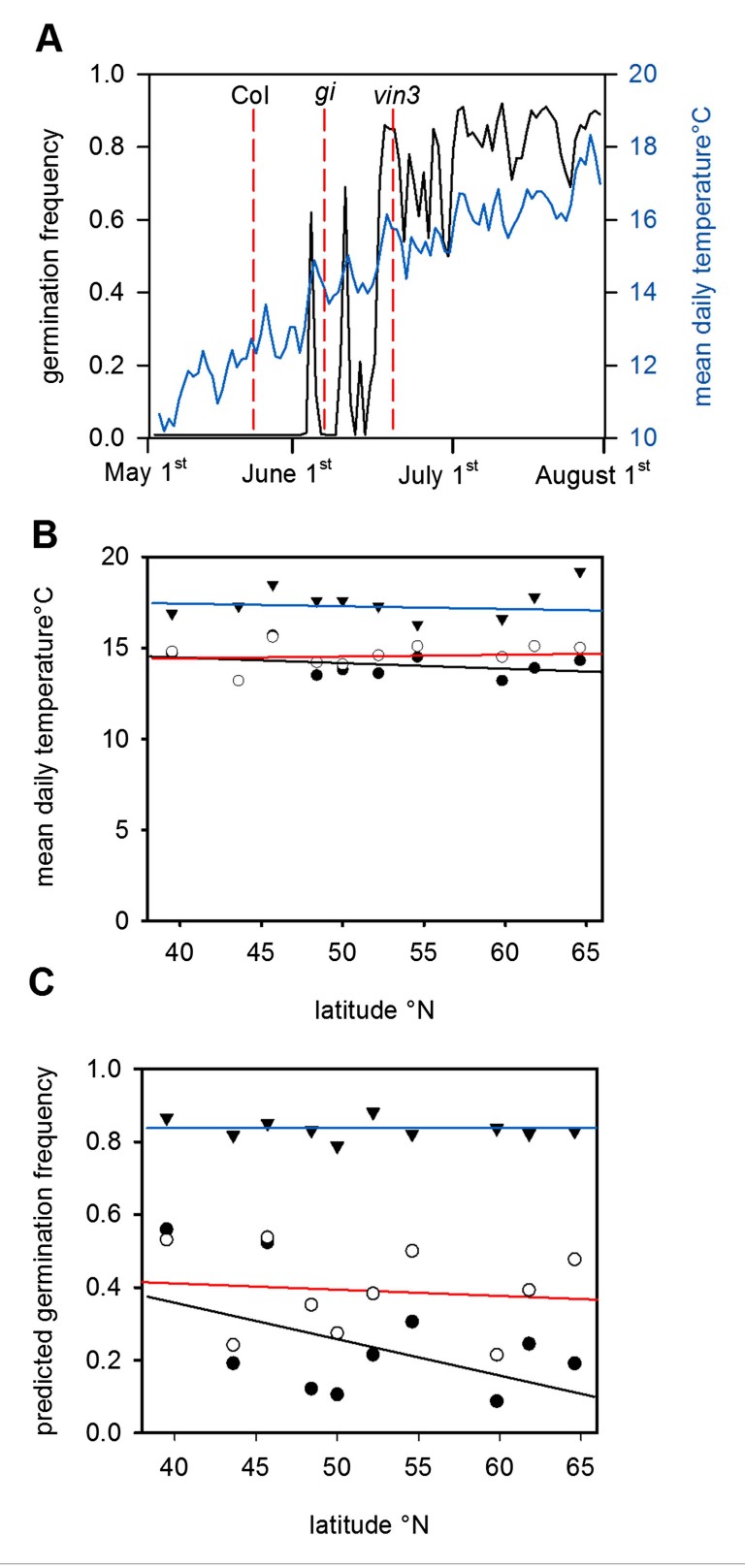

**Figure 4**. The predicted effect of genetic variation in flowering time on seed set temperature and progeny behaviour. (**A**) Time to bolting for the *gi-201* and *vin3-1* mutants lines was calculated using previously published parameter sets (7) for Gorsow, Poland and overlayed on ten year average mean temperature (blue) and model-predicted germination

*Figure 4. continued on next page*

*Figure 4. Continued*

(black), assuming that both *GI* and *VIN3* affect germination only indirectly via flowering time. (**B**) Predicted mean daily temperature at first seed set for winter annual Col-0 (closed circles), *gi* (open circles), and *vin3* (closed triangles) across the European climate transect (*Figure 1D*), using the flowering time and seed set models. Note that sensitivity of life history parameters to GI increases with increasing latitude with wider variation in annual photoperiod. (**C**) Assuming germination is spread evenly over 1 month after first seed set, predicated germination of the whole progeny of Col-0, *gi* and *vin3-1* winter annual seeds across the transect. Later flowering has the potential to lead to large changes in the number of low-dormant seeds due to the extreme sensitivity of seed set to temperature in this range (*Figure 2*).

but seed behaviour and seed bank dynamics are central to long-term population persistence: in Norway the mean lifespan of *Arabidopsis* in the seed bank was estimated at between 1 and 8 years (*Lundemo et al., 2009*), but work to understand genetic contributions to seed bank dynamics is in its infancy.

Col-0 germination in autumn or spring gives a similar flowering date in May. This suggests that Col-0 can operate equally well as a summer or winter annual. This prediction is supported by field emergence data, which shows that seed set in the laboratory at 15˚C or seed set in the field conditions germinate in the soil in both autumn and spring emergence windows (*Figure 6*). This bet-hedging strategy can therefore be relevant to both summer and winter annual accessions. Previously it has been shown that seeds of the Cape Verde Island accession germinate only in an autumn emergence window in central England (*Footitt et al., 2011*), whereas accession Bur-0 shows only a spring emergence window (*Footitt et al., 2013*). However, our work shows that winter and spring annual behaviour does not necessarily require different genotypes and are not mutually exclusive strategies. Both spring and autumn germination windows have also been described in coastal but not montane Spanish populations (*Montesinos et al., 2009*), suggesting that the strategy employed by Col-0 is of wide relevance in some ecological contexts.

In the case of Col-0, winter annual behaviour appears to require (1) a strong slowing of growth rates under low temperatures, showing that growth rate can be as important as leaf number in determining timing of the floral transition in natural populations, and (2) strong tendency towards fast secondary dormancy induction combined with an autumn germination window. In contrast, strong *FRI* and *FLC* alleles and other late flowering alleles also confer flowering delays primarily on summer germinants (*Wilczek and et al., 2009*). This achieves winter annual behaviour with a second mechanism that permits early seed germination. The increased prevalence of strong *FLC* alleles at northern latitudes (*Shindo et al., 2006*; *Li et al., 2014*) is good evidence that an advantage of this latter strategy is maximisation of energy capture in short growing seasons through early seedling establishment, rather than generation of winter annual behaviour itself. At mid-latutides stronger *FRI/FLC* alleles have the potential to push back seed set of summer germinants into autumn (*Wilczek and et al., 2009*), and during autumn the temperature again passes back through the critical 14–15˚C range. This may then stratify the behaviour of progeny seeds of these cohorts.

In addition to flowering time, latitudinal clines in seed dormancy have also been described, whereby accessions form more northerly latitudes exhibit lower primary dormancy levels (*Atwell et al., 2010*). Given that decreasing temperature during seed maturation strongly increases primary dormancy, our analysis suggests that these opposing genetic and environmental trends could offset each other in field conditions allowing maintenance of progeny behavioural strategies. Variation at the *DELAY OF GERMINATION1* (*DOG1*) locus is a major determinant of natural variation in primary seed dormancy (*Bentsink et al., 2006*; *Chiang et al., 2011*) and temperature-regulation of *DOG1* transcript levels during seed maturation (*Chiang et al., 2011*; *Kendall et al., 2011*; *Chiang et al., 2013*) may therefore play an important role in adaptation of seed dormancy to changing temperatures.

Recent articles also show that it is possible to model *Arabidopsis* populations over several generations using thermal time models (*Donohue et al., 2014*; *Burghardt et al., 2015*). The process reveals theoretical interactions between primary dormancy and flowering time that can recreate aspects of *Arabidopsis* phenology observed previously under field conditions at different latitudes, including the ability of Col-0 to adopt winter annual, summer annual, and rapid cycling life histories described here. Our analysis also shows that germination behaviour not included in these models such as temperature regulation of primary dormancy depth and secondary dormancy kinetics are necessary

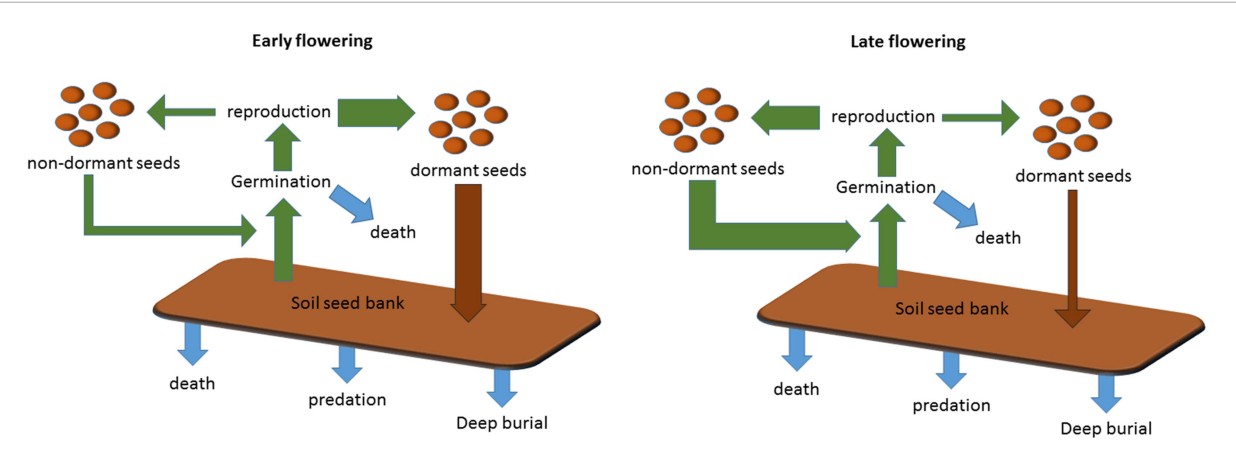

**Figure 5**. Scheme to show the role of winter annual flowering time control in soil seed bank formation. Seed bank persistence requires that seed entry rate (production of dormant seeds) exceeds exit rate, the sum of germination, death or deep burial. Width or arrows indicates relative flux variation with flowering time. Early flowering plants produce larger proportions of dormant seeds, increasing seed bank size. Later flowering plants produce fewer dormant seeds but more rapid cyclers, potentially producing more seeds from the next generation.

to understand life history control. Combining these two complimentary approaches has the potential to enable a new level of understanding of the diversity of life history strategy in annual plants and their genetic basis.

A further feature of Col-0 phenology is that the bet-hedging strategy is stable over latitude throughout Europe, except in the far north where temperature is too cool to generate germinating cohorts. It is also stable in respect to simulated warming (*Figure 3—figure supplement 1*), except in respect to time of year of seed set. This surprising lack of local adaptation could facilitate population stability in changing environments or enable rapid colonization of new territories, if timing of seed set was under no further constraints. Understanding the generality of this strategy is therefore of clear importance. Our work suggests that integration of life history modelling with behaviour of genetic variants has the potential to reveal fitness tradeoffs across the whole life history and identify non-obvious selection pressures as emergent features.

## Materials and methods

### Growth chamber experiments
For seed production, Col-0 plant were grown in sanyo MLR 350 growth cabinets at 80–100 μmol m s$^{-1}$ white light at 22˚C in 16 hr light 8 hr dark cycles until bolting. At bolting, plants were transferred to the indicated seed maturation environment. For temperature manipulations photoperiod was maintained constant at 16 hr light 8 hr dark, and for photoperiod manipulations the temperature remained constant at 22˚C. Seeds were harvested when approximately 50% of siliques had dehisced. Each batch was then sieved through a 250-μm mesh (Fisher Scientific, Basingstoke, UK) to exclude poorly filled seeds. Approximately, 25–50 freshly harvested seeds from each plant were sown onto 0.9% water agar plates, and at least five plants per treatment were used to generate biological replicate seed batches. Plates were wrapped in foil to prevent light from reaching the seeds. A single un-stratified plate was used as the zero time point and the remaining plates were wrapped in foil to exclude light, and incubated in growth cabinets at 4˚C, 8˚C, 12˚C, 15˚C, or 20˚C. After 3, 7, 14, 21, 28, and 42 days, a single plate from each stratification treatment was removed and placed in 12 hr white light dark cycles at 80 μmol m s$^{-1}$ and 22˚C for 7 days to permit germination of non-dormant seeds, which was scored as radical protrusion. Occasionally, seeds had germinated during the dark incubation phase and were easily identified by elongated hypocotyls. It was presumed that these seeds were non-dormant at the start of the experiment and were therefore discarded from further analysis.

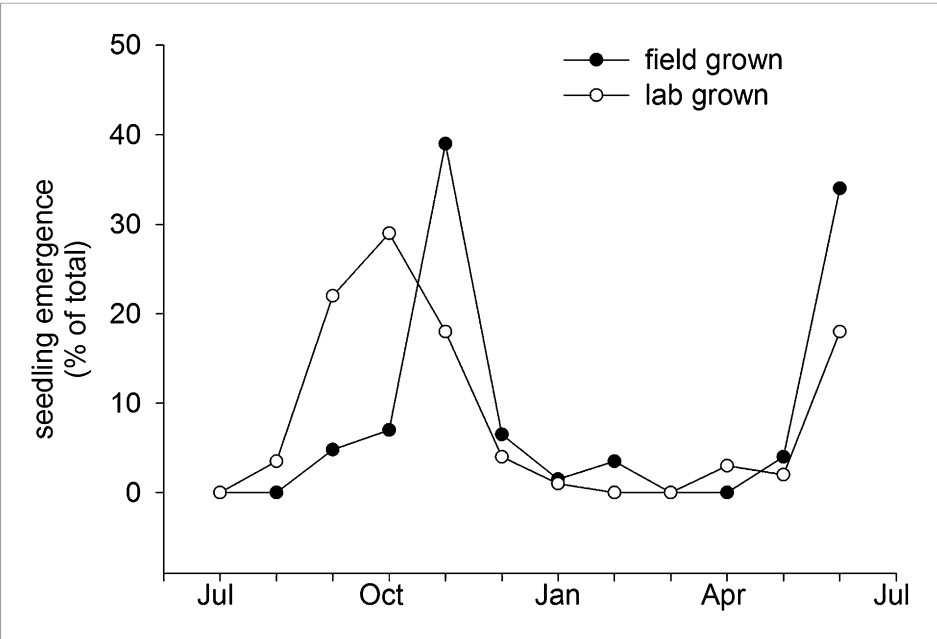

**Figure 6**. Field emergence time in 2013/14 of Col-0 seed set in the laboratory at 15°C or set in the field in York in spring 2013. Data represent the total percentage emergence at 2 weekly intervals of 500 seeds sown for each experiment.

## Field experiments

Five independent growings of plants were raised at a field site between October 2011 and July 2013. Col-0 seeds were germinated at 22°C after stratification at 4°C for 2–3 days, transplanted into John Innes seed compost (Levington) in P40 trays, and kept moist for 1 week in Sanyo cabinet operating at 15°C to harden off. Trays were then moved to the field site, a walled garden within the University of York campus grounds, and dates of bolting, first and last mature seed were recorded for each individual. Slug pellets were applied when necessary, and fences around the perimeter excluded vertebrate herbivores. Seeds from five of these batches were harvested and used for germination time series experiments of the same format as described above. To compare weather station readings with soil level air temperature, LogTag TRIX-8 temperature loggers (LogTag Recorders Ltd, Auckland, New Zealand) were used. Loggers were placed inside trays containing plants in order to shield the sensor from direct sunlight. Loggers were set to record the temperature every 10 min, and mean hourly temperatures were calculated from these readings.

Seedling emergence experiments were performed as described previously (*Footitt et al., 2011*). Briefly, 500 seeds from a mixed batch of lab grown seed set at 15°C or field grown seed in York UK in spring 2013 were sown in June 2013 in pots of John Innes seed compost, and pots were sunk into the ground at our field site at the University of York, UK. Once every two weeks each pot was exhumed, seed and soil spread on a tray and exposed to natural light for 5 min. Germinated seeds were removed and scored before replacing the soil and reburying the pot.

## Climate data origins

Daily average, maximum and minimum temperatures for the period January 2011 to August 2013 were collected from the University of York Department of Electronics weather station archive (see sources, below). The weather station was situated approximately 300 m away from the field site, and approximately 21 m above ground level. This was preferred to soil temperature measurements because it more closely reflects temperature recordings by weather stations used for life history simulations.

An additional 14 locations were selected based on a transect across continental Europe spanning the habitat of natural accessions of *Arabidopsis thaliana* and close to the believed site of collection for Col-0. Temperature data spanning 10 years were collected (www.geodata.com), and mean values for

daily average, daily minimum and daily maximum temperatures were calculated for each day of the year. Times of sunrise and sunset at the nearest available location to each weather station were also collected, however since these do not vary significantly year to year, only 1 year of data was used. Photothermal models require temperature estimates to a resolution of 1 hr. Therefore, estimates of hourly temperature were produced from daily average maximum and minimum temperatures by assuming daily minimum between the hours of 21:00 and 02:00, daily maximum between 09:00 and 14:00, and average daily temperature for the remaining hours. These hourly temperature estimates, alongside daily sunrise and sunset times were used as the basis for simulations.

## Germination model development

A simple model in which the probability that any random seed in a population will germinate $P(G)$ is mutually exclusive to the probability that it is dormant $P(D)$ was defined as:

$$P(G) = 1 - P(D). \tag{1}$$

The idea of simultaneous independent loss of primary dormancy and induction of secondary dormancy (*Totterdell and Roberts, 1979*; *Batlla et al., 2009*) was used to explain germination behaviour in *Arabidopsis* seed populations. The individual probabilities of both primary dormancy $P(D_p)$ and secondary dormancy $P(D_s)$ were considered to be independent, and thus the total probability of dormancy $P(D)$ was derived using the general disjunction rule as follows:

$$P(D) = P(D_p \cup D_s) = P(D_p) + P(D_s) - P(D_p)P(D_s). \tag{2}$$

Population-based threshold models (*Lundemo et al., 2009*) have shown that variation in germination timing could be due to differences in base water potential ($\psi_b$). This parameter is defined as the minimum water potential required for germination and is normally distributed within seed populations. The idea of using $\psi_b$ as a threshold value means that for any environmental water potential ($\psi$), only seeds with $\psi_b$ values exceeding $\psi$ are capable of germination.

Dormancy breaking or inducing treatments have also been shown to alter the mean $\psi_b$ in seed populations. This translates to increasing proportions of seeds either losing or gaining dormancy as the population mean moves in relation to the threshold. If the mean dormancy were to change with a constant rate, a cumulative distribution curve would emerge from plotting these percentages over time. This concept was applied to both primary and secondary dormancy processes. It was assumed that seed populations have initially high mean values of primary dormancy and low mean values of secondary dormancy. Over time the mean primary dormancy would decrease, resulting in a cumulative reduction in the percentage of primary dormant seeds, while the opposite trend was presumed to occur for secondary dormancy.

A cumulative distribution function of dormancy in general could be defined as the probability that a random seed has a dormancy value $D$ less than or equal to the threshold $d$ and can be found by integrating the probability density function, or normal distribution $f(x)$ between the limits $-\infty$ and $d$ and as follows:

$$F(d) = P(D \leq d) = \int_{-\infty}^{d} f(x)\ dx. \tag{3}$$

In this definition, it is the threshold $d$ which varies rather than the population mean; however, the two concepts are mathematically equivalent. Logistic functions were chosen to reproduce the desired S-shape of a cumulative distribution with only a small number of parameters. This approach conveniently negates any need to determine whether seeds are primary or secondary dormant, measure dormancy thresholds, population parameters, or have any knowledge about the agents causing dormancy.

The following equations describe logistic functions which were used to model the probabilities of primary and secondary dormancy over time $x$:

$$P(D_p) = \frac{1}{1 + e^{R_p(x + A_p)}}, \tag{4}$$

$$P(D_s) = 1 - \left[ \frac{1}{1 + e^{R_s(x + A_s)}} \right], \tag{5}$$

where $R_p$ and $R_s$ are the rates of primary dormancy loss and secondary dormancy induction; $A_p$ and $A_s$ are offset parameters. Because a standard logistic curve passes through $x = 0$ at $y = 0.5$, $A_p$ and $A_s$ were chosen such that the curves were repositioned so that $P(D_p)$ crosses $x = 0$ at 0.99 and $P(D_s)$ crosses $x = 0$ at 0.01. The magnitude of the two offset parameters required to position the curves were found by rearranging equations (*Fitter and Fitter, 2002*; *Cotton, 2003*) and then substituting values of $x = 0$ and either $P(D_p) = 0.01$ or $P(D_s) = 0.99$.

Training data were generated by setting Col-0 seeds at 12°C, 13°C, 14°C, 15°C, 17°C and 18°C and incubating seeds at temperatures between 4°C and 20°C for up to 8 weeks in the dark before transferring to the light at the times indicated for germination at 22°C (*Figure 2*). Any seeds that germinated in the dark prior to light exposure were excluded from the analysis. In this data set there appeared a clear trend for an increase in the rate of secondary dormancy induction with higher incubation temperatures.

The model was fitted using the *fit* function of MATLAB and allowed to optimise $R_p$ independently for each germination time-series. This revealed a clear trend such that lowering maturation temperature decreased $R_p$; we therefore produced nine different models consisting of all possible combinations of linear, logistic, and exponential functions to model rates of primary and secondary dormancy loss or induction with maturation and incubation temperature respectively and fitted to the entire training data set using the *fit* function in the MATLAB curve fitting toolbox (Mathworks). The best model based on the fit with the training data was a linear model of primary dormancy loss with maturation temperature ($T_m$) and an exponential model of secondary dormancy induction with incubation temperature ($T_i$) shown in *Equations 6, 7* (*Table 1*) and the fit to data shown in *Figure 2—figure supplement 2*.

$$R_p = aT_m + b, \tag{6}$$

$$R_s = ce^{dT_i}. \tag{7}$$

Combining *Equations 1, 2* with *Equations 4–7* results in a conceptually simple probabilistic model of germination with only 4 parameters ($a,b,c,d$) which determine the rates of primary dormancy loss and secondary dormancy induction with temperature. Optimised parameter values are shown in *Table 2*.

## Model validation

A good fit was achieved by optimising model parameters using the training data (*Figure 2—figure supplement 2*); however, a better test of the model is to assess its ability to predict data not used in

**Table 1**. Comparisons of $R^2$ and parameter number of the nine germination models tested with linear, exponential or logistic relationship between $R_p$ and $R_s$ with temperature

| $R_p$ model | $R_s$ model | Number of parameters | Total $R^2$ |
|---|---|---|---|
| Exponential | Exponential | 4 | 0.71 |
| Exponential | Logistic | 5 | 0.62 |
| Exponential | Linear | 4 | 0.64 |
| Logistic | Exponential | 5 | 0.83 |
| Logistic | Logistic | 6 | 0.82 |
| Logistic | Linear | 5 | 0.75 |
| Linear | Exponential | 4 | 0.83 |
| Linear | Logistic | 5 | 0.81 |
| Linear | Linear | 4 | 0.73 |

**Table 2**. Optimised final parameter values for the germination and seed set models

| Parameter | Value |
|---|---|
| Germination model | |
| A | 1.56 |
| B | −21.79 |
| C | 0.05 |
| D | 0.18 |
| Seed set model | |
| Tb (base temperature) | 5.25 |
| Threshold | 5370 |

its construction. Data obtained from maturing seeds at 16°C were used as a validation test (*Figure 2—figure supplement 3*). This produced a good fit to data ($R^2 = 0.88$) and therefore the model could predict the germination of seed lots produced and incubated at different temperatures in the lab not used to train the data.

To understand whether the model was successful at predicting germination in seed lots from real field situations, the model was fitted against data sets generated from five field-grown populations (*Figure 2—figure supplement 4*). The parameters and incubation temperatures were either fixed or known, and the MATLAB *fit* function was used to derive a predicted value for $T_m$ for each lot which could be compared to temperature data logged from the field sites between flowering and seed set (*Figure 2—figure supplement 4*). In each case, the predicted $T_m$ appeared close to that during the final few days of seed set, suggesting that this period is critical in determining dormancy depth in *Arabidopsis*, as previously determined in sorghum.

## Seed set model development

Photothermal time models integrate temperature and photoperiod information over time using a function such as the one below (*Borthwick et al., 1943*; *Thomas and Raper, 1976*).

$$T_h = \int_a^b f(x) \, dx. \tag{8}$$

Photothermal model of seed development rate over time $x$, is integrated from the point of bolting $a$, up to the point of first seed maturity, $b$. This permits the calculation of $T_h$, the total thermal time experience required to achieve the production of mature seeds. In variable environments, it is necessary to approximate the above integral with a summation of the function over discrete time intervals. The summation function becomes as follows:

$$T_h = \sum_{t=1}^{n} \theta(t), \tag{9}$$

where $\theta(t)$ is the thermal time accumulated at time $t$, and $T_h$ is the cumulative total between bolting at $t = 1$, and seed maturity $t = n$. If $T_h$ is known in advance, this concept can be used to determine the value of $n$, and hence the time required to produce mature seeds in any particular environment. To develop the best model of developmental rates of Col-0 seed set was measured in multiple temperature regimes (*Figure 1*) and photoperiod regimes (*Figure 1—figure supplement 1*). The thermal time model was as follows:

$$\theta(t) = \begin{cases} (T(t) - Tb), & T(t) \geq Tb \\ 0, & T(t) < Tb \end{cases}, \tag{10}$$

where $\theta(t)$ is the number of thermal time units accumulating during the hour beginning at time $t$, $T(t)$ is the average temperature during hour $t$, and $Tb$ is the genotype-specific base temperature. We also compared a photothermal time model:

$$\theta(t) = \begin{cases} (T(t) - Tb) \times P(t), & T(t) \geq Tb \\ 0, & T(t) < Tb \end{cases}, \tag{11}$$

where $\theta(t)$ is the number of photothermal time units for the hour beginning at time $t$, $T(t)$ is the average temperature (°C) during the time interval, and $Tb$ is the base temperature, which is a genotype-specific constant. $P(t)$ is a measure of daylight.

In order to parameterise the models seed was set under laboratory conditions at a range of temperatures between 8°C and 25°C, and at photoperiods between 8 hr light periods and 16 hr light periods (*Figure 1A*). Optimal parameters were calculated using the *fit* function of MATLAB. The performance of each model was evaluated by comparing the predictions of seed development times with nine independent recorded seed development times in the field. These data were collected between October 2011 and July 2013 and closely matched the predictions for each model. The photothermal model ($R^2 = 0.28$) did not outperform the simple thermal time model ($R^2 = 0.42$): therefore, we elected to use a simple thermal time model of *Arabidopsis* seed set (*Figure 1*).

## Acknowledgements

This work was supported by a BBSRC studentship to VS and a Royal Society University Research Fellowship to SP. The authors would like to thank Ozgur Akman for useful advice in the construction of the Col-0 seed germination model.

## Additional information

### Funding

| Funder | Grant reference | Author |
| --- | --- | --- |
| Biotechnology and Biological Sciences Research Council (BBSRC) | | Vicki Springthorpe |
| Royal Society | University Research Fellowship | Steven Penfield |

The funders had no role in study design, data collection and interpretation, or the decision to submit the work for publication.

### Author contributions

VS, Acquisition of data, Analysis and interpretation of data, Drafting or revising the article; SP, Conception and design, Analysis and interpretation of data, Drafting or revising the article

### Author ORCIDs

Vicki Springthorpe, http://orcid.org/0000-0002-7430-3712

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
