## [Decision Letter]

Thank you for sending your work entitled “Flowering time and seed dormancy control use external coincidence to generate life history strategy” for consideration at *eLife*. Your article has been favorably evaluated by Detlef Weigel (Senior editor) and 2 reviewers, one of whom, Richard Amasino, is a member of our Board of Reviewing Editors. The other reviewer, Maarten Koornneef, has also agreed so share his identity.

The Reviewing editor and the other reviewer discussed their comments before we reached this decision, and the Reviewing editor has assembled the following comments to help you prepare a revised submission.

Your paper on flowering time and seed dormancy is likely to be acceptable for publication if you address the following:

It would be useful to readers of your work if you provide additional information on why you chose data from [39] on the flowering time of the *gi* and *vin3* mutants to incorporate into your modeling. Such mutants are unlikely to proliferate in nature, and the most interesting question is how the range of natural variation in flowering might affect, for example, the ratios of dormant to non-dormant seeds in a changing climate. Thus, it would be interesting to elaborate a bit on natural variation.

We are not asking for you to do additional experiments, but rather to discuss the above more thoroughly so that your work is placed in a broader context.

Indeed, it is understandable that the experimental work is done only for the lab strain Col and not for multiple genotypes that differ also in dormancy levels. The importance of genetic variation in dormancy levels is addressed by several papers of the Donohue group and it would be useful when the authors in their discussion can speculate how genetic variation would affect their model; e.g., with respect to being summer or winter annual. The outcome of the present study is that Col, which is normally considered to have the attributes of a summer annual (early flowering, low dormancy in most lab conditions), can be winter or summer annual, which in Western Europe seems relatively rare.

Additional experiments on genetic variation in relation to life cycle have been published by Footitt and Finch-Savage, as well as by Pico and co-workers. It would be useful to discuss the bandwidth due to genetic differences for this species that seems to have its flowering time in nature well controlled (in Western and Central Europe from spring to early summer) despite strong variation observed in lab and greenhouse experiments for the traits that seem to matter.

[Editors' note: further revisions were requested prior to acceptance, as described below.]

Thank you for resubmitting your work entitled “Flowering time and seed dormancy control use external coincidence to generate life history strategy” for further consideration at *eLife*. Your revised article has been favorably evaluated by Detlef Weigel (Senior editor) and the two original reviewers.

The manuscript has been improved but there are some remaining issues that need to be addressed before acceptance, as outlined below:

The Discussion has been improved in your revision. However, to make your paper most useful for and accessible to a general audience (a goal of *eLife*), we ask that you elaborate on the relationship of your work to other recently published work. Specifically, two papers appeared after your initial submission ([14], Trends Ecol. Evol., and [7], The American Naturalist). These papers describe a somewhat similar approach to predict life history over the year in different environments. The similarities and differences in the approaches and resulting models in your work to that of another group will not be clear for the average (non-modeler) reader without further discussion.

One difference is that the other group includes genetic variation both in flowering time and dormancy in their models. In the previous round of review, a reviewer asked you to address the consequences of genetic variation. This was addressed for flowering time, but not for dormancy. As both your paper and that of another group are likely to have a strong impact on the field, it would be useful for you to devote a paragraph in the Discussion to comparing models.

---

## [Author Response]

*It would be useful to readers of your work if you provide additional information on why you chose data from*
[39]
*on the flowering time of the* gi *and* vin3 *mutants to incorporate into your modeling. Such mutants are unlikely to proliferate in nature, and the most interesting question is how the range of natural variation in flowering might affect, for example, the ratios of dormant to non-dormant seeds in a changing climate. Thus, it would be interesting to elaborate a bit on natural variation*.

*We are not asking for you to do additional experiments, but rather to discuss the above more thoroughly so that your work is placed in a broader context*.

We have expanded the justification for using the *gi* and *vin3* parameter sets, which are also analysed at length in the Wilczek 2009 paper. The reason we chose them was that *gi* confers are very moderate (7–10 days) delay in field conditions (and not in winter), and *vin3* confers a more substantial delay (20–50 days) all year round. This allows us to analyse the effect of delaying flowering time in general (by any allele, known or unknown, natural or man-made) as well as the effect of both loss-of-functions, which we agree are unlikely to proliferate in nature. Available parameter sets from natural alleles such as *FRI/FLC* do not affect flowering of spring flowering winter annuals (because *FLC* is vernalised away) so it was not informative to consider them in this analysis because they do not cause a delay (at least in the Wilczek 2009 paper parameter sets and field data). Known natural variants of FT and CO may be worth considering but parameter sets for introgressed Col–0 NILs would be required for this analysis. For summer behaviour see response to comment below.

*Indeed, it is understandable that the experimental work is done only for the lab strain Col and not for multiple genotypes that differ also in dormancy levels. The importance of genetic variation in dormancy levels is addressed by several papers of the Donohue group and it would be useful when the authors in their discussion can speculate how genetic variation would affect their model; e.g., with respect to being summer or winter annual. The outcome of the present study is that Col, which is normally considered to have the attributes of a summer annual (early flowering, low dormancy in most lab conditions), can be winter or summer annual, which in Western Europe seems relatively rare*.

*Additional experiments on genetic variation in relation to life cycle have been published by Footitt and Finch-Savage as well as by Pico and co-workers. It would be useful to discuss the bandwidth due to genetic differences for this species that seems to have its flowering time in nature well controlled (in Western and Central Europe from spring to early summer) despite strong variation observed in lab and greenhouse experiments for the traits that seem to matter*.

Some good points here. We have added our field-emergence time data for Col–0, which was obtained using methods similar to those used by Footitt and Finch-Savage (see Methods for details). This confirms your view that Col–0 has both a spring and a summer germination window. This has also been observed in coastal Spanish populations ([27], now cited and discussed), but in that case it was not clear if the seeds germinating in the two windows were genetically identical (they were just dug from soil seed banks). We agree that it is useful to discuss the cited work and have included all this in a substantial new Discussion section (paragraphs 3 and 4), and the field emergence data as Figure 6.

[Editors' note: further revisions were requested prior to acceptance, as described below.]

*The Discussion has been improved in your revision. However, to make your paper most useful for and accessible to a general audience (a goal of* eLife*), we ask that you elaborate on the relationship of your work to other recently published work. Specifically, two papers appeared after your initial submission (*[14]*, Trends Ecol. Evol., and*
[7]*, The American Naturalist). These papers describe a somewhat similar approach to predict life history over the year in different environments. The similarities and differences in the approaches and resulting models in your work to that of another group will not be clear for the average (non-modeler) reader without further discussion*.

*One difference is that the other group includes genetic variation both in flowering time and dormancy in their models. In the previous round of review, a reviewer asked you to address the consequences of genetic variation. This was addressed for flowering time, but not for dormancy. As both your paper and that of another group are likely to have a strong impact on the field, it would be useful for you to devote a paragraph in the Discussion to comparing models*.

We agree that this work is relevant and have included the citations and a comment. Basically the difference is that these model populations of 1000 plants use only a highly stylised description of seed germination behaviour, with only genetically determined primary dormancy and no secondary dormancy. Our analysis and that of [17] show that environmental control of dormancy depth and secondary dormancy are in fact key responses that determine field germination behaviour, and the behaviour we discuss is only revealed if these are correctly treated. However, combining the two approaches has clear merit, a point which we have emphasised. The two papers do reach one common conclusion that Col–0 is theoretically capable of all known *Arabidopsis* life histories, and this point I have emphasized in the text. In our case this conclusion is confirmed by field data.

In addition, we have expended to include a discussion of genetic variation in dormancy depth. This is interesting and I think we now understand one aspect of this, that lower dormancy observed in accessions from Northern latitudes in the lab likely functions to offset the effects of colder environments increasing primary dormancy levels. We have discussed the likely roles of natural variation at the DOG1 locus in this apparent adaptive response.